# In Vitro Antimicrobial Activity of *Nymphaea pubescens* (Pink Water Lily) Leaf Extracts

**DOI:** 10.3390/plants12203588

**Published:** 2023-10-16

**Authors:** Boontarika Thongdonphum, Kittima Vanichkul, Adun Bunchaleamchai, Pannapa Powthong

**Affiliations:** 1Faculty of Agricultural Technology, Rajamangala University of Technology Thanyaburi, Thanyaburi 12130, Thailand; kittima_v@rmutt.ac.th; 2Faculty of Medical Technology, Rangsit University, Mueang Pathum Thani 12000, Thailand; adun.b@rsu.ac.th; 3Faculty of Science, Rangsit University, Mueang Pathum Thani 12000, Thailand; pannapa.p@rsu.ac.th

**Keywords:** *Nymphaea pubescens*, phytochemical, antimicrobial activity

## Abstract

This research comparatively investigates the in vitro antimicrobial activity of extracts from *Nymphaea pubescens* (pink water lily) leaves against pathogenic bacteria. The experimental extracts are aqueous, acetonic, and 95% ethanolic *N. pubescens* extracts; and the pathogenic bacteria being studied include *Aeromonas hydrophila*, *Vibrio parahaemolyticus*, *Vibrio vulnificus*, and *Vibrio harveyi*, which are commonly found in freshwater fish and brackish aquatic animals. The ethanolic *N. pubescens* extract achieves the highest bacterial inhibitory effects against *V. parahaemolyticus* and *V. vulnificus*. The minimum inhibitory concentrations of the ethanolic extract against *A. hydrophila* and *V. harveyi* are 10 mg/mL; and 2.5 mg/mL against *V. parahaemolyticus* and *V. vulnificus*. The ethanolic *N. pubescens* extract is effective against *V. parahaemolyticus*. The high-performance liquid chromatography results show that, in the phenolic acids group, gallic acid is the most dominant (0.600–3.21% *w*/*w*), followed by sinapic acid (0.37–0.83% *w*/*w*). In the flavonoids group, catechin is the most dominant (0.02–1.08% *w*/*w*), followed by rutin (0.002–0.03% *w*/*w*). Essentially, the ethanolic *N. pubescens* extract can potentially be used as a natural antibiotic agent to treat bacterial infections in fish and aquatic animals.

## 1. Introduction

Antimicrobial residues and contaminants in aquatic ecosystems are closely linked to the misuse and overuse of antimicrobials to treat infectious diseases in aquatic animals [1]. In addition to being harmful to humans and the environment, antimicrobial residues can cause antimicrobial resistance in microbes. Specifically, antimicrobial resistance occurs when microbes evolve mechanisms that protect them from the effects of antimicrobials [2].

Antimicrobials are medicines used to prevent and treat infections in humans, animals, and plants. They include antibiotics, antivirals, antifungals, and antiparasitics. Of particular interest are antibiotics which are widely used in the prevention and treatment of bacterial infections. In Thailand, the use of antibiotics in animal husbandry and aquaculture has been projected to increase by 67%, or from 63,000 tons in 2010 to 106,000 tons in 2030 [3]. The overuse of antibiotics, however, contributes to the prevalence of antibiotic-resistant bacteria or superbugs [4], which in turn are the primary sources of mobile genetic elements of fish pathogens [5].

The literature contained studies on the antimicrobial properties of plant extracts belonging to the family Nymphaeaceae [4,6,7]. *Nymphaea* (water lily) is a genus of hardy and tender aquatic plants in the family Nymphaeaceae. Water lilies are medicinal plants with a wide range of therapeutic effects, including aphrodisiac, anodynic, astringent, cardiotonic, sedative, analgesic, and anti-inflammatory properties [6]. The plants in the genus *Nymphaea* have traditionally been used to treat infections and diseases [8]. A comprehensive review of more than one hundred scientific studies was conducted and the findings showed that the content of phenolics and flavonoids of the *Nymphaea* medicinal plants varied, depending on the plant parts, e.g., flowers, leaves, stems, roots, and rhizomes [9].

*Nymphaea pubescens* (pink water lilies) have traditionally been used to treat a wide range of illnesses. *N. pubescens* is rich in phytochemicals, including flavonoids, alkaloids, phenolic acids, terpenoids, anthraquinones, saponins, and tannins [7]. Alkaloids have bacterial inhibitory effects against infectious diseases, while phenolic acids and tannins have bacterial inhibitory effects against pathogenic microorganisms. Phenolic acids possess antimicrobial, anti-inflammatory, and anti-cancer properties [10].

The pathogenic bacteria commonly found in freshwater fish and brackish aquatic animals include *Aeromonas hydrophila*, *Vibrio parahaemolyticus*, *Vibrio vulnificus*, and *Vibrio harveyi*. The genera *Aeromonas* and *Vibrio* are the causative agents of bacterial infections in freshwater fish and brackish aquatic animals. Specifically, *A. hydrophila* is the main causative agent of infectious diseases in freshwater fish [11]. This bacterial species causes fin or basal hemorrhage, skin and gill lesions, abdominal distension, cutaneous necrotic ulceration, and erosion of infected fish’s fins and orbicularis [12,13,14].

*V. parahaemolyticus* causes acute hepatopancreatic necrosis disease (AHPND) in *Penaeus vannamei* shrimps (whiteleg shrimps). The shrimp infected with AHPND exhibits lethargy, anorexia, slow growth, an empty digestive tract, and a pale-to-white hepatopancreas [15]. *V. parahaemolyticus* is a leading cause of gastrointestinal illness in humans if ingested [16]. *V. vulnificus* is commonly found in marine environments such as estuaries, brackish ponds, or coastal areas. This bacterial species is one of the causative agents of white feces disease (WFD) in Pacific white shrimp (*Litopenaeus vannamei*) [17]. WFD is a disease that attacks the shrimp’s digestive system. *V. harveyi* causes eye lesions/blindness, gastro-enteritis, muscle necrosis, skin ulcers, and tail rot disease in marine fish. This bacterial species is also the etiological agent of luminous vibriosis in shrimps [18].

The emergence of antibiotic-resistant bacteria in aquaculture due to excessive use of antibiotics necessitates the development of more potent synthetic (chemically based) antibiotics to control the spread of the superbugs. However, more potent antibiotics are more harmful to human and aquatic animal health and the environment. As a result, this research proposes, as an alternative to synthetic antibiotics, plant-based extracts with bacterial inhibitory effects. Unlike the chemically based antibiotics, the plant-derived extracts are more environmentally friendly and safer for human and aquatic animal health. Furthermore, the plant-based extracts are less likely to cause antibiotic resistance in bacteria.

Specifically, this research experimentally extracts *Nymphaea pubescens* (pink water lily) leaves using distilled water (aqueous), acetone, and 95% ethanol. The in vitro antimicrobial activity of the extracts against the pathogenic bacteria commonly found in freshwater fish and brackish aquatic animals are subsequently investigated. The pathogenic bacteria being studied include *A. hydrophila*, *V. parahaemolyticus*, *V. vulnificus*, and *V. harveyi*. The antimicrobial activity of the aqueous, acetonic and ethanolic extracts of *N. pubescens* leaves are determined in terms of the minimum inhibitory concentration and the minimum bactericidal concentration. Moreover, high-performance liquid chromatography is used to identify and quantify the phenolic and flavonoid compounds in the *N. pubescens* extracts. Essentially, this research is the first to experimentally extract phytochemical compounds from *N. pubescens* leaves, and the first to study the antimicrobial activity of the extracts against the pathogenic bacteria commonly found in freshwater fish and brackish aquatic animals.

## 2. Experimental Results and Discussion

Table 1 tabulates the screening results of the phytochemicals in the *N. pubescens* extracts. The aqueous extract of *N. pubescens* leaves contained flavonoids, alkaloids, phenolic acids, terpenoids, anthraquinones, saponins, and tannins. The phytochemicals present in the acetonic extract were alkaloids, phenolic acids, and tannins, while the ethanolic extract of *N. pubescens* leaves contained flavonoids, alkaloids, phenolic acids, and tannins.

The preliminary screening results showed that the *N. pubescens* leaves contained several major groups of phytochemicals, including flavonoids, alkaloids, phenolic acids, terpenoids, anthraquinones, saponins, and tannins. This is consistent with [19], who reported the presence of alkaloids, phenolic acids, and tannins in water lilies. Alkaloids are effective against infectious diseases, while phenolic acids and tannins have bacterial inhibitory effects against pathogenic microorganisms. Phenolic acids also possess antimicrobial, anti-inflammatory, and anti-cancer properties [10]. Alkaloids, flavonoids, and saponins in the ethanolic and acetonic extracts possess antibacterial properties and antioxidant capacity [20]. Meanwhile, the content of phenolics and flavonoids of the *Nymphaea* medicinal plants varied widely, depending on the plant parts, e.g., flowers, leaves, stems, roots, and rhizomes [9].

Table 2 compares the inhibition zone (in vitro antimicrobial activity) of the *N. pubescens* leaf extracts at 10 mg/mL concentration, 10% DMSO, and oxytetracycline (100 µg/mL) against *A. hydrophila*, *V. parahaemolyticus*, *V. vulnificus*, and *V. harveyi*. The aqueous extract failed to inhibit the pathogenic bacteria, while the acetonic and ethanolic extracts exhibited the inhibitory effects against *V. parahaemolyticus* and *V. vulnificus*. The experiments showed that the ethanolic *N. pubescens* extract achieved the highest bacterial inhibitory effects against *V. parahaemolyticus* and *V. vulnificus*, as evidenced by the largest inhibition zones (14.7 ± 0.6 mm for *V. parahaemolyticus*, and 12.7 ± 0.6 mm for *V. vulnificus*).

In Table 2, the results showed that 10% DMSO failed to inhibit the bacteria. Although oxytetracycline was more effective in inhibiting the pathogenic bacteria than the acetonic and ethanolic extracts, the *N. pubescens* leaf extracts are more environmentally friendly and less harmful to human and aquatic animal health [2]. Moreover, the *N. pubescens* extracts are also less likely to cause antibiotic resistance in bacteria [8]. Figure 1a–d shows the inhibition zone of the *N. pubescens* leaf extracts (10 mg/mL), 10% DMSO, and oxytetracycline against the experimental pathogenic bacteria.

Table 3 presents the MIC of the *N. pubescens* extracts against the experimental pathogenic bacteria. The results showed that the inhibitory effects of the aqueous, acetonic, and ethanolic extracts against *A. hydrophila* and *V. harveyi* were identical, as indicated by the MIC of 10 mg/mL. The ethanolic *N. pubescens* extract was effective in inhibiting the growth of *V. parahaemolyticus* and *V. vulnificus*, as evidenced by the lowest MIC (2.5 mg/mL), followed by the acetonic extract with the MIC of 5 mg/mL. Meanwhile, the aqueous extract is less effective in inhibiting the growth of the pathogenic bacteria (MIC = 10 mg/mL).

Table 4 presents the MBC of the *N. pubescens* extracts against the experimental pathogenic bacteria. The results showed that the ethanolic *N. pubescens* extract was very effective in killing the bacterial species *V. parahaemolyticus*, as evidenced by the lowest MBC of 2.5 mg/mL. However, the ethanolic extract was not effective in eliminating *A. hydrophila, V. vulnificus*, and *V. harveyi*, as indicated by MBC ≥ 10 mg/mL. Similarly, the aqueous and acetonic *N. pubescens* extracts were not effective in destroying the pathogenic bacteria, with MBC ≥ 10 mg/mL.

Table 5 tabulates the HPLC results of the aqueous, acetonic, and ethanolic extracts of *N. pubescens* leaves, consisting of six phenolic acids (i.e., caffeic acid, gallic acid, ferulic acid, p-coumaric acid, sinapic acid, and syringic acid) and six flavonoids (apigenin, catechin, kaempferol, myricetin, quercetin, and rutin).

In the phenolic acids group, gallic acid in the aqueous extract was highest (3.21% *w*/*w*), followed by the ethanolic extract (3.08% *w*/*w*) and the acetonic extract (0.6% *w*/*w*). Meanwhile, sinapic acid in the ethanolic extract was highest (0.83% *w*/*w*), followed by the aqueous extract (0.54% *w*/*w*) and the acetonic extract (0.37% *w*/*w*). The HPLC chromatograms showed that the *N. pubescens* extracts contained phenolic acids of variable quantities, depending on the extraction solvents. The phenolic acids present in the *N. pubescens* extracts included caffeic acid, gallic acid, p-coumaric acid, sinapic acid, and syringic acid.

The flavonoids present in the *N. pubescens* extracts were catechin and rutin, with varying quantities depending on the extraction solvents. In the flavonoids group, the ethanolic extract of *N. pubescens* leaves had the highest content of catechin (1.08% *w*/*w*), followed by the aqueous extract (0.17% *w*/*w*) and the acetonic extract (0.02% *w*/*w*). Similarly, the ethanolic extract had the highest content of rutin (0.03% *w*/*w*), followed by the aqueous extract (0.01% *w*/*w*) and the acetonic extract (0.002% *w*/*w*). Non-toxic and biodegradable ethanol is efficient in extracting phytochemicals from plants due to their polarity [21]. Figure 2a–d shows the HPLC chromatograms of the stock solutions, aqueous extract, acetonic extract, and ethanolic extract of *N. pubescens* leaves, respectively.

Specifically, in the phenolic acids group, gallic acid was the most dominant (0.600–3.21% *w*/*w*), followed by synapic acid (0.37–0.83% *w*/*w*). In the flavonoids group, catechin was the most dominant (0.02–1.08% *w*/*w*), followed by rutin (0.002–0.03% *w*/*w*) (Table 5). Gallic acid possesses remarkable antioxidant properties [10]. The ethanolic extract of *N. nouchali* seeds contained 0.27% *w*/*w* of gallic acid [4]. The ethanolic *N. nouchali* extract exhibited the inhibitory effects against negative bacteria, positive bacteria, and pathogens. Sinapic acid is present in fruits, vegetables, seed oils, herbs, and particularly abundant in the *Cruciferae* family. Sinapic acid is widely recognized for its antimicrobial, anti-inflammatory, anti-cancer, and antioxidant properties [22].

Caffeic acid is a hydroxycinnamic acid compound commonly found in wines [23]. Caffeic acid possesses antioxidant properties that enhance immunity, regulate blood lipid levels, and counteract the effects of mutagens. This acid is also found in various fruits, vegetables, and herbs [10]. The *N. lotus* flower extract exhibited minimal toxicity despite administering the extract over an extended period of time and in excessive quantities [24,25]. Due to the minimal toxicity, the plant-derived extracts are more environmentally friendly and safer to human and aquatic animal health. Furthermore, the plant extracts are less likely to cause antibiotic resistance in bacteria.

The *N. pubescens* extracts (i.e., aqueous, acetone, and 95% ethanol) contain the major phytochemical groups with bacterial inhibitory effects, including flavonoids, alkaloids, phenolic acids, terpenoids, anthraquinones, saponins, and tannins. The in vitro antimicrobial assay showed that the *N. pubescens* extracts, particularly the ethanolic *N. pubescens* extract, are effective in inhibiting the growth of *A. hydrophila*, *V. parahaemolyticus*, *V. vulnificus*, and *V. harveyi*, which are the pathogenic bacteria commonly found in freshwater and brackish aquatic animals. Specifically, the ethanolic *N. pubescens* extract is effective in inhibiting the growth of *V. parahaemolyticus* and *V. vulnificus* (MIC = 2.5 mg/mL). The ethanolic *N. pubescens* extract is also effective in eliminating the bacterial species *V. parahaemolyticus* (MBC = 2.5 mg/mL). By comparison, oxytetracycline (i.e., positive control) is more effective in inhibiting the pathogenic bacteria than the *N. pubescens* extracts. The *N. pubescens* leaf extracts are nonetheless less harmful to human and aquatic animal health and are less likely to cause antibiotic resistance in bacteria.

## 3. Materials and Methods

### 3.1. Experimental Plant

The fresh leaves were first thoroughly rinsed with water and oven-dried (Forced Convection Oven-FD115, Binder, Tuttlingen, Germany) at 60 °C for 48 h [26]. The dried *N. pubescens* leaves were ground into powder for subsequent extraction. The authenticity of fresh *N. pubescens* leaves was verified by the Lotus Museum of the Rajamangala University of Technology Thanyaburi.

### 3.2. Preparation of Plant Extracts

In the extraction, the dried *N. pubescens* powder was separately soaked in distilled water (aqueous), acetone, and 95% ethanol. Specifically, 30 g of powdered *N. pubescens* was soaked in 600 mL solvents (i.e., distilled water, acetone, and 95% ethanol) at 30 °C for 48 h on an orbital shaker. The mixture was filtered using Whatman No. 1 filter paper (Whatman, Buckinghamshire, UK), and the filtrate was concentrated using a rotary evaporator (HED-1560-01300-00, Heidolph, Schwabach, Germany) at 50 °C.

### 3.3. Experimental Pathogenic Bacteria

The experimental pathogenic bacteria included *A. hydrophila*, *V. parahaemolyticus*, *V. vulnificus*, and *V. harveyi*, which are four pathogenic bacteria commonly found in freshwater fish and brackish aquatic animals. *A. hydrophila* is a heterotrophic, Gram-negative, rod-shaped bacterium mainly found in freshwater fish, while *Vibrio parahaemolyticus*, *V. vulnificus*, and *V. harveyi* are curved, rod-shaped, Gram-negative bacteria found in brackish aquatic animals, such as whiteleg shrimp. The experimental pathogenic bacteria were inoculated on tryptic soy agar (TSA) and incubated at 37 °C for 24 h (VWR incubator model 1552, Shel lab, Cornelius, OR, USA). *A. hydrophila*, *V. parahaemolyticus*, and *V. vulnificus* were obtained from Thailand’s National Institute of Health, Mueang Nonthaburi, Thailand, and *V. harveyi* was acquired from the Thailand Institute of Scientific and Technological Research, Khlong Luang, Thailand.

### 3.4. Phytochemical Screening of Plant Extracts

A phytochemical screening was carried out with the *N. pubescens* extracts (i.e., aqueous, acetonic, and ethanolic *N. pubescens* extracts) to identify the chemical constituents in the extracts. The main chemical constituents included phenolic acids, flavonoids, tannins, saponins, alkaloids, anthraquinones, and terpenoids. The screening methods followed [27] for phenolic acids, [28] for flavonoids, [29] for tannins, [30] for saponins, [31] for alkaloids, [32] for anthraquinones, and [29] for terpenoids.

### 3.5. Determination of Bacterial Growth Inhibition

The agar well diffusion method was used to determine the bacterial growth inhibition of the aqueous, acetonic, and ethanolic *N. pubescens* extracts [33]. The experiments were carried out using petri dishes containing Mueller–Hinton agar (MHA) spread with the inoculums (i.e., the experimental pathogenic bacteria) using a sterile swab.

Prior to the agar well diffusion assay, the concentrations of the *N. pubescens* extracts in 10% dimethyl sulfoxide (DMSO) were varied between 0.1, 0.5, 1.0, 5.0, and 10 mg/mL. The preliminary results indicated that the concentrations below 10 mg/mL (<10 mg/mL) achieved unsatisfactory inhibition of bacterial growth. As a result, the concentration of 10 mg/mL was used in the agar well diffusion assay.

In the agar well diffusion assay, 30 µL of the *N. pubescens* extract in 10% DMSO (10 mg/mL) was pipetted (Eppendof research micropipette, Eppendof, Selangor Darul Ehsan, Malaysia) onto the wells, which were cut 6.0 mm in diameter. The petri dishes with the pipetted were then incubated at 37 °C for 24 h (VWR incubator model 1552, Shel lab, Cornelius, OR, USA). Afterwards, the inhibition zone of the aqueous, acetonic, and ethanolic *N. pubescens* extracts against the experimental pathogenic bacteria were measured by a caliper. The experiments were carried out in triplicate. In the disc diffusion assay, 10% DMSO was used as negative control and oxytetracycline (100 µg/mL) as positive control.

### 3.6. Determination of Minimum Inhibitory Concentration and Minimum Bactericidal Concentration

The resazurin-based 96-well plate microdilution method [34] was used to determine the minimum inhibitory concentration of the *N. pubescens* extracts against the experimental pathogenic bacteria. The minimum inhibitory concentration (MIC) is defined as the lowest concentration of an antimicrobial extract that inhibits the visible growth of a microorganism after overnight incubation.

Prior to the MIC analysis, *A. hydrophila* was cultured on TSA while *Vibrio* spp. was cultured on TSA and 1.5% NaCl. The cultures (*A. hydrophila*, *V. parahaemolyticus*, *V. vulnificus*, and *V. harveyi*) were inoculated at 37 °C for 24 h. Afterwards, the cultures were subcultured to standard glass tubes (16 mm in diameter) and diluted with 0.85% of NaCl until the bacterial suspension was 0.5 McFarland unit (DEN-1B McFarland densitometer, Biosan, Riga, Latvia). The bacterial suspensions were then diluted 100 times in Mueller–Hinton Broth to a density of 1 × 10^6^ CFU/mL.

In the extraction, the broth microdilution method was used, whereby the *N. pubescens* extracts of 10 mg/mL initial concentration were two-fold serially diluted using Mueller–Hinton Broth (MHB) in 96-well microtiter plates. The diluted concentrations of the *N. pubescens* extracts were 10 (initial), 5, 2.5, 1.25, 0.62, 0.31, 0.15, and 0.08 mg/mL.

In the MIC analysis, 50 µL of the *N. pubescens* extracts of variable concentrations (10, 5, 2.5, 1.25, 0.62, 0.31, 0.15, 0.08 mg/mL) were pipetted onto sterile 96-well microtiter plates. Afterwards, 50 µL of the bacterial suspensions (*A. hydrophila*, *V. parahaemolyticus*, *V. vulnificus*, and *V. harveyi*) were added and mixed well. MHB (100 µL) and 10% DMSO (100 µL) were used as the positive control, and the bacterial suspensions (100 µL) were used as the negative control. The microtiter plates were subsequently incubated at 37 °C for 24 h. The MIC assays were performed in triplicate. After the first incubation, 10 µL of 0.18% resazurin indicator (blue color) was dropped in each well and further incubated at 37 °C for 1 h. After the second incubation, the colorimetric changes were determined, where the blue color indicates the inhibition of bacterial growth (i.e., positive result) and the change from blue to pink color indicates no inhibition of bacterial growth (negative result).

Meanwhile, the minimum bactericidal concentration (MBC) is defined as the lowest concentration of an antimicrobial agent that kills a microorganism (e.g., bacteria). In the MBC analysis, the extract concentrations with positive MIC results were transferred from the 96-well microtiter plates using sterile loop to the petri dishes containing TSA for *A. hydrophila* and TSA and 1.5% NaCl for *Vibrio* spp. The petri dishes were incubated at 37 °C for 24 h, and the MBC were subsequently determined. The MBC assays were carried out in triplicate. Specifically, the MBC was the concentrations of the extracts without bacterial colonies. In the MBC analysis, the bacterial suspensions were used as positive control, while TSA and 10% DMSO were used as negative control.

### 3.7. Determination of Phenolic Acids and Flavonoids

The high-performance liquid chromatography (HPLC) analysis was performed using Agilent technologies 1260 infinity (Conquer Scientific, Poway, CA, USA) equipped with Inertsil C18 column (5 mm, 4.6 × 250 mm) at 35 °C and 272 nm wavelength [10]. The gradient elution was acetonitrile and 1% acetic acid. The flow rate was 1.0 mL/min, and the injection volume was 20 µL. The phytochemical compounds being studied were the phenolic acids group (caffeic acid, gallic acid, ferulic acid, p-coumaric acid, sinapic acid, and syringic acid) and the flavonoids group (apigenin, catechin, kaempferol, myricetin, quercetin, and rutin).

The standard phenolic acids and flavonoids were acquired from Sigma-Aldrick (St. Louis, MO, USA). The HPLC-grade solvents, including distilled water, acetone, and 95% ethanol, were purchased from Merck (Darmstadt, Germany). Stock solutions of the phenolic acids and flavonoids groups were prepared by following [10]. The HPLC detection limits of the stock solutions of phenolic acids and flavonoids groups were 5, 10, 20, 30, 40, and 60 µg/mL, given R^2^ ≥ 0.99. The stock solutions and the *N. pubescens* extracts (i.e., aqueous, acetonic, and ethanolic extracts) at 10 mg/mL concentration were filtered by 0.45 um PVDF syringe filter and the mobile phase was degassed before the injection of the solutions.

## 4. Conclusions

This research investigated the in vitro antimicrobial activity of the extracts from *N. pubescens* leaves against pathogenic bacteria. The *N. pubescens* extracts included aqueous, acetone, and 95% ethanol extracts; and the experimental pathogenic bacteria were *A. hydrophila*, *V. parahaemolyticus*, *V. vulnificus*, and *V. harveyi*, which are commonly found in freshwater fish and brackish aquatic animals. The experimental results showed that the ethanolic *N. pubescens* extract was most effective in inhibiting bacterial growth. The ethanolic extract achieved the highest bacterial inhibitory effects against *V. parahaemolyticus* (14.7 ± 0.6 mm inhibition zone) and *V. vulnificus* (12.7 ± 0.6 mm). The MIC of the ethanolic extract against *A. hydrophila* and *V. harveyi* were 10 mg/mL; and 2.5 mg/mL against *V. parahaemolyticus* and *V. vulnificus*, respectively. The ethanolic extract of *N. pubescens* leaves was also very effective in eliminating *V. parahaemolyticus*. The HPLC results indicated that, in the phenolic acids group, gallic acid (0.600–3.21% *w*/*w*) and synapic acid (0.37–0.83% *w*/*w*) were the most and second most dominant active components. In the flavonoids group, catechin was the most dominant (0.02–1.08% *w*/*w*), followed by rutin (0.002–0.03% *w*/*w*). Although oxytetracycline was more effective in inhibiting the pathogenic bacteria than the acetonic and ethanolic extracts, the *N. pubescens* leaf extracts are more environmentally friendly and less harmful to human and aquatic animal health. Moreover, the *N. pubescens* extracts are also less likely to cause antibiotic resistance in bacteria. Essentially, the ethanolic *N. pubescens* extract holds promising potential as a natural antibiotic agent (i.e., an alternative to synthetic antibiotics) to prevent and treat bacterial infections in freshwater fish and brackish aquatic animals. As a result, subsequent research would experimentally mix the ethanolic *N. pubescens* extract with fish meal to treat infectious diseases in aquatic animals.

## Figures and Tables

**Figure 1 plants-12-03588-f001:**
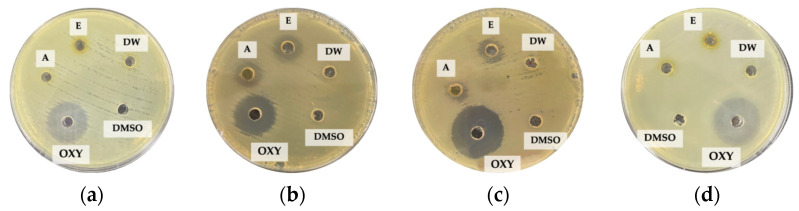
The inhibition zone of *N. pubescens* leaf extracts, 10% DMSO and oxytetracycline against: (**a**) *A. hydrophila*, (**b**) *V. parahaemolyticus*, (**c**) *V. vulnificus*, (**d**) *V. harveyi*, where DW = aqueous extract, A = acetonic extract, E = ethanolic extract, DMSO = 10% DMSO, and OXY = oxytetracycline.

**Figure 2 plants-12-03588-f002:**
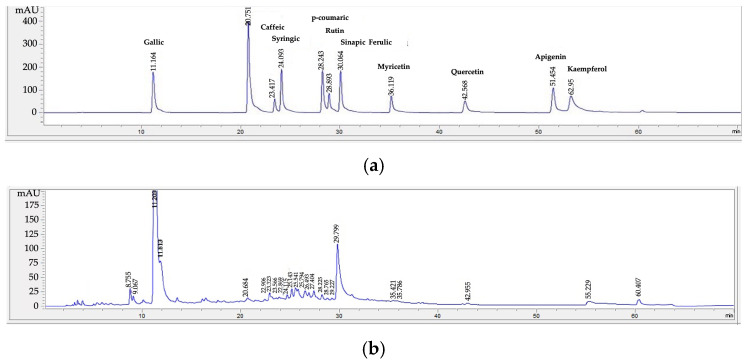
The HPLC chromatograms showing the presence of phenolic acids and flavonoids: (**a**) the stock solutions, (**b**) aqueous extract, (**c**) acetonic extract, (**d**) ethanolic extract of *N. pubescens* leaves.

**Table 1 plants-12-03588-t001:** The major phytochemical groups in *N. pubescens* leaf extracts.

Phytochemicals	Aqueous Extract	Acetone Extract	Ethanol Extract
Flavonoids	+	-	+
Alkaloids	+	+	+
Phenolic acids	+	+	+
Terpenoids	+	-	-
Anthraquinones	+	-	-
Saponins	+	-	-
Tannins	+	+	+

Note: + denotes the presence of the compound and - the absence of the compound.

**Table 2 plants-12-03588-t002:** The zone of inhibition (antimicrobial activity) of the experimental extracts from *N. pubescens* leaves (10 mg/mL), 10% DMSO (negative control), and oxytetracycline (positive control).

Extract/Control	Zone of Inhibition (mm ± SD)			
	*A. hydrophila*	*V. * *parahaemolyticus*	*V. vulnificus*	*V. harveyi*
Aqueous extract	No	No	No	No
Acetone extract	No	12.3 ± 0.6	11.0 ± 1.0	No
Ethanol extract	No	14.7 ± 0.6	12.7 ± 0.6	No
10% DMSO	No	No	No	No
OXY	22.7 ± 0.6	20.0 ± 0.0	25.3 ± 0.6	25.0 ± 0.0

Note: OXY denotes oxytetracycline (100 µg/mL), and No indicates no inhibition of the pathogenic bacteria.

**Table 3 plants-12-03588-t003:** The minimum inhibitory concentration (MIC) of the extracts from *N. pubescens* leaves against the experimental pathogenic bacteria.

*N. pubescens* Extract	MIC (mg/mL)			
	*A. hydrophila*	*V. parahaemolyticus*	*V. vulnificus*	*V. harveyi*
Aqueous extract	10	10	10	10
Acetone extract	10	5	5	10
Ethanol extract	10	2.5	2.5	10

**Table 4 plants-12-03588-t004:** The minimum bactericidal concentration (MBC) of the extracts from *N. pubescens* leaves against the experimental pathogenic bacteria.

*N. pubescens* Extract	MBC (mg/mL)			
	*A. hydrophila*	*V. parahaemolyticus*	*V. vulnificus*	*V. harveyi*
Aqueous extract	>10	10	10	>10
Acetone extract	>10	10	10	>10
Ethanol extract	>10	2.5	10	>10

**Table 5 plants-12-03588-t005:** The HPLC results of the aqueous, acetonic and ethanolic extracts of *N. pubescens* leaves.

Phytochemicals	% *w*/*w* (mean ± SD)		
	Aqueous Extract	Acetone Extract	Ethanol Extract
**Phenolic acids**			
Caffeic acid	0.02 ± 0.00	0.02 ± 0.00	-
Gallic acid	3.21 ± 0.00	0.60 ± 0.00	3.08 ± 0.00
Ferulic acid	-	-	-
p-coumaric acid	0.02 ± 0.00	0.02 ± 0.00	0.05 ± 0.00
Sinapic acid	0.54 ± 0.00	0.37 ± 0.00	0.83 ± 0.01
Syringic acid	0.01 ± 0.00	0.10 ± 0.00	0.77 ± 0.01
**Flavonoids**			
Apigenin	-	-	-
Catechin	0.17 ± 0.00	0.02 ± 0.00	1.08 ± 0.01
Kaempferol	-	-	-
Myricetin	-	-	-
Quercetin	-	-	-
Rutin	0.01 ± 0.00	0.002 ± 0.00	0.03 ± 0.00

Note: - denotes not detected.

## Data Availability

The article encompasses all the data that support the findings of the study.

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
