# Peer review of "In Vitro Antimicrobial Activity of Nymphaea pubescens (Pink Water Lily) Leaf Extracts"

_plants, 2023, doi:10.3390/plants12203588_

Round 1

Reviewer 1 Report

Fonts are very inconsistent through out the paper. Please fix this issue and there are extra spaces in the manuscript. 

Reviewer 2 Report

Confirmation of the presence of groups of biologically active compounds in the analysed plant material does not bring much and does not allow to draw the right conclusions. Such methods are not used in modern research. The title of the publication includes the terms "phytochemical composition", while in the paper only the content of selected phenolic compounds was assessed. Therefore, it cannot be ruled out that other compounds, the presence of which has only been confirmed by unreliable tests, are responsible for the biological activity of the raw material, e.g. the alkaloids mentioned in the literature review.

The whole article, especially the introduction and the discussion of the results, are chaotic and do not always refer to the problem under study. In the text - both in the introduction and in the discussion, there is no continuity of argument/thought. The work requires significant corrections both in relation to the results and the presented text.

Author Response

Response to Reviewer’s Comments

Title: In Vitro Antimicrobial Activity of Nymphaea pubescens (Pink Water Lily) Leaf Extracts

Date of Revision Submission: 18/9/2023

Comment 1: Confirmation of the presence of groups of biologically active compounds in the analyzed plant material does not bring much and does not allow to draw the right conclusions. Such methods are not used in modern research.

Answer/Action: In this research, the phytochemical screening was primarily used to identify the major phytochemical groups present in the N. pubescens extracts. The preliminary screening confirmed the presence of the following major phytochemical groups: phenolic acids, flavonoids, tannins, saponins, alkaloids, anthraquinones, and terpenoids.

However, this current research focused on the phenolic acids and flavonoids groups because the phenolic compounds and flavonoids have been reported to possess antimicrobial capacity against pathogenic bacteria (Sun, 2023). Besides, both phenolics and flavonoids are the most and second most dominant bioactive compounds in N. lotus (Daffodil and Mohan, 2013; Semaming et al, 2018; Tungmunnithum et al,  2022).

In response to your comment/suggestion, subsequent research would investigate the antimicrobial activity of the remaining phytochemical groups (i.e., tannins, saponins, alkaloids, anthraquinones, and terpenoids), in addition to the phenolic acids and flavonoids groups. Moreover, the future research would also assess the toxicity (i.e., half maximal inhibitory concentration or IC50) of the extracts in fish.  

References:

Sun, W.; Shahrajabian, M.H. Therapeutic Potential of Phenolic Compounds in Medicinal Plants-Natural Health Products for Human Health. Molecules, 2023, 28, 1845. 

Daffodil, E.D.; Mohan, V.R. Total Phenolics, Flavonoids and In Vitro Antioxidant Activity of Nymphaea pubescens Wild Rhizome. World Journal of Pharmacy and Pharmaceutical Sciences, 2013, 2(5), 3710-3722.

Semaming, Y. Antioxidant Activity and Protective Effect Against Oxidative Stress Induced Hemolysis of Nymphaea lotus L. Extracts. Asia-Pacific Journal of Science and Technology, 2018, 23(4), 1-7.

Tungmunnithum, D.; Drouet, S.; Garros, L.; Hano, C.  Differential Flavonoid and Other Phenolic Accumulations and Antioxidant Activities of Nymphaea lotus L. Populations throughout Thailand. Molecules, 2022, 27, 3590.

Comment 2: The title of the publication includes the terms "phytochemical composition", while in the paper only the content of selected phenolic compounds was assessed. Therefore, it cannot be ruled out that other compounds, the presence of which has only been confirmed by unreliable tests, are responsible for the biological activity of the raw material, e.g. the alkaloids mentioned in the literature review.

Answer/Action: As per your comment/suggestion, the research title has been revised to accurately capture the scope of this research work. The new research title is as below:

In Vitro Antimicrobial Activity of Nymphaea pubescens (Pink Water Lily) Leaf Extracts

To your comment that other active compounds could be responsible for the biological activity of the extracts, the subsequent research would therefore determine the antimicrobial activity of the remaining phytochemical groups, in addition to the phenolic acids and flavonoids groups.

Comment 3: The whole article, especially the introduction and the discussion of the results, are chaotic and do not always refer to the problem under study. In the text - both in the introduction and in the discussion, there is no continuity of argument/thought. The work requires significant corrections both in relation to the results and the presented text.

Answer/Action: As per your comment, the manuscript has been revised in its entirety for ease of comprehension. In addition, the organization of the paper has been rearranged to do away with confusion.

Reviewer 3 Report

Dear authors,

Please could you check and replace synaptic acid line 21, 226 and 385 by synapic acid. Enhance the figure 1 that we could see what is write on the photo! 

You have described the activity of the extract and argue that 

 The N. pubescens leaf extracts are nonetheless less harmful to human and aquatic animal health and are less likely to cause antibiotic resistance in bacteria. 

A Toxicity test of your extract against a normal cell could be of great interest to evaluate the specific index of your extracts.

Is it just the mix of Phenolic acids (gallic acid well known to have antibacterial activity but Cinnamic acid have a MIC of 6.75mM against V. parahaemolyticus molecules 2019 also) That give the activity? 

Rutin has no activity against V. parahaemolyticus Food Control 2021.

Author Response

Response to Reviewer’s Comments

Title: In Vitro Antimicrobial Activity of Nymphaea pubescens (Pink Water Lily) Leaf Extracts

Date of Revision Submission: 18/9/2023

Comment 1: Please could you check and replace synaptic acid line 21, 226 and 385 by synapic acid. Enhance the figure 1 that we could see what is write on the photo! 

Answer/Action: The erroneous words “synaptic acid” have been changed to “synapic acid” in the revised manuscript accordingly. Besides, improvements have been made to the descriptions in Figure 1 for legibility.

Comment 2: You have described the activity of the extract and argue that 

The N. pubescens leaf extracts are nonetheless less harmful to human and aquatic animal health and are less likely to cause antibiotic resistance in bacteria. A Toxicity test of your extract against a normal cell could be of great interest to evaluate the specific index of your extracts.

Answer/Action: The focus of this current research is to investigate the in vitro antimicrobial activity of extracts from Nymphaea pubescens (pink water lily) leaves against pathogenic bacteria. Nonetheless, in response to your suggestion, our subsequent research would also assess the toxicity (i.e., half maximal inhibitory concentration or IC50) of the extracts in fish fed with fish meal mixed with the N. pubescens extracts. 

Comment 3: Is it just the mix of Phenolic acids (gallic acid well known to have antibacterial activity but Cinnamic acid have a MIC of 6.75mM against V. parahaemolyticus molecules 2019 also) That give the activity?  Rutin has no activity against V. parahaemolyticus Food Control 2021.

Answer/Action: The antibacterial activity of the N. pubescens extracts could be attributed to other phytochemical compounds (including cinnamic acid), in addition to phenolic acids and flavonoids.  Nonetheless, the focus of this current research is the phenolic acids and flavonoids groups since both phytochemical groups have been reported to possess antimicrobial capacity against pathogenic bacteria (Sun, 2023). Furthermore, both phenolics and flavonoids are the most and second most dominant bioactive compounds in N. lotus (Daffodil and Mohan, 2013; Semaming et al, 2018; Tungmunnithum et al,  2022). In response to your comment/suggestion, our subsequent research would therefore look into the antimicrobial activity of other phytochemical groups, in addition to the phenolic acids and flavonoids groups.

To another comment of yours, Lui et al (2022) studied the minimum inhibitory concentration (MIC) of rutin against the bacterial species V. pahaemolyticus and reported the MIC of 83.71% given 100 ppm of rutin.

Reference:

Liu, J.; Tong, J.; Wu, Q.; Liu, J.; Yuan, M.; Tian, C.; Xu, H.; Pradeep K. Malakar, P.K.; Pan, Y.; Zhao, Y.; Zhang, Z. Natural Inhibitors Targeting the Localization of Lipoprotein System in Vibrio parahaemolyticus. Int. J. Mol. Sci. 2022, 23, 14352.
